# Association between proton pump inhibitor use and upper gastrointestinal cancer: A matched case-control study accounting for reverse causation and confounding by indication

Ibrahim O. Sawaid[1], Zohar Din[2], Efrat Golan[3], Eytan Ruppin[4,5], Avivit Golan-Cohen[2,6], Ilan Green[2,6], Eugene Merzon[2,7], Shlomo Vinker[2,6], Abraham O. Samson[1], Ariel Israel[2,8]*

**1** Drug Discovery Lab, Azrieli Faculty of Medicine, Bar-Ilan University, Safed, Israel, **2** Department of Family Medicine and Leumit Research Institute, Leumit Health Services, Tel-Aviv, Israel, **3** Pediatric Intensive Care Unit, Haemek Medical Center, Afula, Israel, **4** Cancer Data Science Laboratory, National Cancer Institute, Bethesda, Maryland, United States of America, **5** Translational Research Institute and the Department of Surgery, Cedars-Sinai Medical Center, Los Angeles, California, United States of America, **6** Department of Family Medicine, Grey Faculty of Medical & Health Sciences, Tel-Aviv University, Tel-Aviv, Israel, **7** Adelson School of Medicine, Ariel University, Ariel, Israel, **8** Department of Epidemiology and Preventive Medicine, Gray Faculty of Medical & Health Sciences, Tel-Aviv University, Tel-Aviv, Israel

* dr.ariel.israel@gmail.com

## Abstract

### Background

Proton pump inhibitors (PPIs) are widely used for acid-related disorders, but observational studies have raised concerns about a possible association between long-term PPI use and upper gastrointestinal (GI) cancers. These associations may reflect confounding by indication and reverse causation. We aimed to evaluate the association between PPI use and upper GI cancer while explicitly addressing these biases.

### Methods and findings

We conducted a matched case-control study using electronic health records from a national health organization in Israel. Cases were 875 adults (age 63.0 ± 11.9 years, 62.5% male) with incident upper GI cancer (esophageal, gastric, or duodenal) diagnosed between 2003 and 2024; each case was matched to 10 cancer-free controls (*n* = 8,750). Matching was performed on age, sex, ethnic sector (general, Jewish ultra-orthodox, and Arab), socioeconomic status, and year of enrollment. PPI exposure was ascertained from pharmacy records and modeled in discrete pre-diagnosis windows (0–6 months, 6–12 months, 1–3 years, and 3–10 years). Multivariable conditional logistic regression estimated adjusted odds ratios (aORs) and confidence intervals (CIs), with covariates including age, smoking, body mass index, socioeconomic status, healthcare utilization, pregnancy history (in women), alcohol

**Data availability statement:** The individual-level electronic health record data underlying this study contain potentially identifiable health information and cannot be shared publicly due to ethical, legal, and contractual restrictions imposed by data-protection regulations in Israel. De-identified data may be made available to qualified researchers upon reasonable request, contingent on approval by the Leumit Health Services Institutional Review Board (IRB) and execution of a data-use agreement. Requests for access should be directed to the Leumit Health Services IRB Secretariat (apincu@leumit.co.il). The study authors are not members of the data access committee and do not have special access privileges; external researchers will be granted access on the same terms following the IRB's review. The statistical analysis was performed using *R*. The code used in the analysis is available from Github https://github.com/arielisr/upper_gi_models and archived in Zenodo https://doi.org/10.5281/zenodo.17155577.

**Funding:** This research was internally funded by Leumit Health Services. E.R. is supported by the Intramural Research Program, National Institutes of Health, National Cancer Institute, and Center for Cancer Research. The funders had no role in considering the study design or in the collection, analysis, interpretation of data, writing of the report, or decision to submit the article for publication.

**Competing interests:** The authors have declared that no competing interests exist.

use, *Helicobacter pylori* diagnosis, and upper GI symptom-related diagnoses (e.g., gastroesophageal reflux, gastritis, peptic ulcer disease).

In models without adjustment for symptom-related diagnoses, PPI use was associated with increased odds of cancer (e.g., esomeprazole aOR 4.01, 95% CI 3.20, 5.03, $p < 0.001$; omeprazole aOR 2.38, 95% CI 1.99, 2.85, $p < 0.001$). When exposure was modeled by time window, associations diminished for exposures >1 year before diagnosis. After excluding the final year before diagnosis and adjusting for symptom-related diagnoses, we did not detect a harmful association between PPI use and upper GI cancer. Remote use (>3 years) was instead associated with lower odds (e.g., omeprazole aOR 0.62, 95% CI 0.51, 0.75, $p < 0.001$), with similar patterns in a gastric-only subgroup (701 cases, 7,010 controls). Key limitations include potential residual confounding, lack of direct dietary and family-history data, and incomplete capture of over-the-counter PPI use.

## Conclusions

Apparent harmful associations between PPI use and upper GI cancer were concentrated in the months immediately preceding diagnosis and disappeared after adjusting for diagnostic context and excluding the final year before diagnosis. In these adjusted analyses, we found no evidence of increased odds with long-term PPI use, and remote use (>3 years before diagnosis) was associated with reduced cancer odds for omeprazole and lansoprazole. These findings underscore the importance of investigating new-onset upper GI symptoms rather than attributing malignancy risk to acid-suppressive therapy.

## Author summary

### Why was this study done?

- Proton pump inhibitors (PPIs) are common medicines that reduce stomach acidity.

- Some studies suggest that long-term PPI use might increase the odds of cancers in the upper digestive tract (stomach, esophagus, and small intestine).

- We wanted to test whether PPIs are truly linked to these cancers after accounting for the timing of use and for digestive symptoms that might lead to PPI prescriptions.

### What did the researchers do and find?

- We analyzed medical records from Israel, comparing 875 people with upper GI cancer to 8,750 similar people without cancer, matched on key characteristics.

- We examined PPI purchases in several time windows before diagnosis and adjusted for common digestive conditions and symptoms.

- PPI use within 6 months before diagnosis appeared linked to higher cancer odds, but after adjustment for symptoms, PPI use was not associated with increased odds; instead, use more than 3 years earlier was associated with reduced odds (for omeprazole and lansoprazole).

## What do these findings mean?

- The apparent link between PPIs and cancer may reflect people starting PPIs because of early cancer symptoms (reverse causation), rather than the medicines causing cancer.

- When the timing of PPI use and underlying digestive conditions are taken into account, long-term PPI use is not associated with increased odds of upper GI cancer.

- Main limitations: our purchase data cover about 10 years pre-diagnosis (very long-term effects might be missed), over-the-counter PPI use and some factors like detailed diet or family-history were not fully captured, and results may not generalize to all countries or populations.

## Introduction

Proton pump inhibitors (PPIs) are widely used in the treatment of acid-related gastrointestinal (GI) disorders [1]. Since the introduction of omeprazole in 1988, several PPIs—such as lansoprazole, pantoprazole, rabeprazole, esomeprazole, and dexlansoprazole [2]—have become mainstays in the management of gastroesophageal reflux disease (GERD), peptic ulcer disease, Zollinger–Ellison syndrome, and *Helicobacter pylori* infections [3]. PPIs irreversibly inhibit the gastric $H^+/K^+$-ATPase, leading to prolonged suppression of gastric acid secretion [4].

Before the advent of PPIs, histamine-2 receptor blockers (H2RBs), such as famotidine and ranitidine, were the primary agents for acid suppression [5]. However, the reversible nature of H2RBs binding and the development of tachyphylaxis limited their long-term efficacy [6]. In contrast, PPIs offer superior and sustained acid suppression, which has contributed to their widespread use and, in many cases, prolonged duration of therapy.

While the short-term safety and efficacy of PPIs are well-established, observational studies and meta-analyses have raised concerns about potential long-term harms, including chronic kidney disease [7], cardiovascular events [8], fractures [9], infections [10], and dementia [11]. Several studies, including meta-analyses, have also reported an increased risk of gastric cancer with long-term PPI use [12–20], particularly following *H. pylori* eradication, intensifying debate about whether chronic acid suppression might promote carcinogenesis [21]. Proposed mechanisms center on PPI-induced hypergastrinemia with trophic effects on the gastric mucosa and acid-related shifts in the gastric microbiome leading to mucosal proliferation that may facilitate carcinogenic processes [16,21–25]. Consistent with these hypotheses, long-term acid suppression in rats has been associated with enterochromaffin-like (ECL) cell hyperplasia and gastric neuroendocrine tumors [26], and experience with potent H2 receptor blockade supports a gastrin-mediated trophic pathway under chronic hypoacidity [27].

Nevertheless, the association between PPI use and gastric cancer remains controversial [28,29]. Patients prescribed PPIs often have underlying gastrointestinal symptoms or disorders (for example, dyspepsia, reflux disease, or peptic ulcer) that are themselves associated with increased cancer risk. As a result, observational studies can be susceptible to confounding by indication and protopathic bias, particularly when diagnostic context and the timing of exposure are not explicitly modeled [20,30]. In such settings, PPI use may function as a marker of early disease or its risk factors rather than a causal agent. Several studies have reported that adjusting for these conditions attenuates or eliminates the association [31], while others have shown that elevated risk estimates cluster shortly before diagnosis, consistent with reverse causation or confounding by indication [32]. Together, these studies suggest that the apparent association between PPI use and gastric cancer observed in some analyses could reflect reverse causation or confounding by indication rather than a

true causal relationship. However, the extent to which these biases account for the reported associations remains uncertain, and our study was specifically designed to address this question by modeling exposure timing and diagnostic context.

In this study, we evaluated the association between PPI use and upper GI cancer using a matched case-control design. Importantly, we modeled drug exposure across multiple time windows prior to diagnosis, allowing us to test whether observed associations differ according to timing. We further adjusted for a range of GI diagnoses that may reflect early cancer symptoms or predisposing conditions. By disentangling medication effects from underlying disease processes, we aimed to clarify whether PPIs independently contribute to cancer risk—or whether their association reflects confounding by indication or reverse causation.

## Methods

### Study design and data source

We conducted a retrospective matched case-control study using electronic health records (EHRs) from Leumit Health Services (LHS), one of Israel's four national health providers. LHS provides care to approximately 730,000 members and maintains a centralized longitudinal EHR system comprising over two decades of data on demographics, diagnoses, medication prescriptions and pharmacy purchases, laboratory tests, and healthcare utilization.

All Israeli residents are entitled to universal healthcare under a standardized benefits package, and medication coverage is determined by a national formulary. Diagnoses in LHS are entered by treating physicians using International Classification of Diseases (ICD) codes. The accuracy of diagnostic coding in the LHS registry is considered high and has supported multiple publications in leading peer-reviewed journals [33,34].

### Study population and matching

The study included all LHS members with active enrollment between 2003 and 2024. Cases were defined as individuals aged 18–80 years with a diagnosis of upper GI malignancy, identified through the Israeli National Cancer Registry and LHS diagnostic codes. The earliest cancer diagnosis date served as the index date. Individuals with a prior cancer diagnosis were excluded. During the study period, Leumit Health Services used ICD-9 for clinical coding, whereas the Israeli National Cancer Registry applied ICD-10. To ensure complete ascertainment, we merged both sources and used the earliest record from either system as the index diagnosis date. This dual coding minimized potential under-capture of cancer diagnoses. Of the 892 detected upper GI cancer cases, 680 (76%) were identified via linkage to the Israeli National Cancer Registry, which provided structured data on tumor site, morphology, and histological grade (see S2–S4 Tables) in ICD-10 coded form. The remaining cases were identified through validated ICD-9 codes in the LHS medical record system (ICD-9 codes beginning with 150–152; S5 Table). Among registry-verified cases, the most common anatomical sites were gastric antrum (n = 93), cardia (n = 85), and body of stomach (n = 61). The most frequent morphologies were adenocarcinoma (n = 317), signet ring carcinoma (n = 104), and squamous cell carcinoma (n = 36). Poorly differentiated tumors (grade III) were the most prevalent (n = 275), followed by moderately differentiated tumors (grade II, n = 121).

Cases and controls were matched at a 1:10 ratio based on sex, ethnic sector (general population, Ultra-Orthodox Jewish, or Arab), socioeconomic status, and year of enrollment in LHS. Among eligible controls, individuals with the closest possible date of birth and no history of cancer were selected. The 1:10 ratio was selected to optimize statistical power given the relatively low incidence of upper GI cancer in the cohort and the availability of a large eligible control population. A higher number of controls per case improves the precision of effect estimates while preserving representativeness and limiting selection bias. Each case was matched to exactly 10 cancer-free controls; cases for which 10 exact matches on the specified variables could not be identified were excluded. Post-matching balance on key baseline covariates—including age, sex, ethnic sector, smoking status, alcohol-related diagnoses, and Body Mass Index (BMI)—was evaluated using standardized mean differences and paired p-values, with p > 0.05 considered indicative of acceptable balance.

## Sociodemographic classification

Socioeconomic status was derived from residential address using the Points Location Intelligence system, which ranks neighborhoods on a 1–20 scale aligned with the socioeconomic status indicators defined by the Israeli Central Bureau of Statistics. For analysis, values were grouped into six ordinal categories: 1–3, 4–6, 7–9, 10–11, 12–14, and ≥15. These groupings are consistent with prior national health research and allow for adequate discrimination across the socioeconomic gradient. Ethnic sector classification was based on residential clustering algorithms validated against census data in prior national studies [35]. The algorithm identifies ethnic sector based on neighborhood demographic composition; ethnicity is not directly recorded in Israeli EHRs.

## Medication and symptom exposure assessment

Medication exposures were determined from pharmacy purchase records covering up to 10 years prior to the index date. Medications were categorized using Anatomical Therapeutic Chemical (ATC) classification codes. In addition to PPIs, we included medications commonly used in our health system to treat upper GI symptoms, namely famotidine (a histamine H2 receptor antagonist) and antacids (e.g., calcium carbonate). Exposure was assessed within predefined time windows: 0–0.1, 0.1–0.5, 0.5–1, 1–2, 2–3, and 3–10 years before the index date. For each drug and time window, exposure was defined as a binary variable, set to positive if at least one purchase occurred during the interval.

Upper GI symptoms were identified from medical visits during the same 10-year period using ICD-9 diagnostic codes. Symptom categories included dyspepsia, gastroesophageal reflux, abdominal pain, gastritis, peptic ulcer disease, and *H. pylori* infection, with the list of ICD-9 codes detailed in S6 Table. *H. pylori* status was captured from physician-assigned diagnoses recorded before the index date. Where available, entries reflected endoscopic or histologic confirmation and/or noninvasive testing documented in the clinical record. Because ascertainment relied on coded diagnoses rather than systematic testing, these records may underrepresent lifetime *H. pylori* exposure.

## Covariates

All multivariable models adjusted for demographic and lifestyle variables, including age; sex; smoking status (non-smoker, past smoker, and current smoker); socioeconomic status (SES), derived from residential address and grouped into six ordinal categories 1–3, 4–6, 7–9, 10–11, 12–14, and ≥15 on a 1–20 national scale); body mass index (BMI, categorical; cutoffs: < 18.5, 18.5–25 [reference], 25–30, 30–35, 35–40, 40–45, 45–50, 50–55, ≥ 55 kg/m$^2$); number of physician visits in the years before the index date; healthcare worker status; and pregnancy history in women. For BMI and SES, a "missing" category was used for individuals with missing data (<10%). Models additionally adjusted for GI diagnoses/diseases, including documented *H. pylori* diagnosis, alcohol use diagnoses, GERD, gastritis, peptic ulcer disease, abdominal pain, and constipation. Information on dietary intake and family-history of cancer was not available in the EHR; BMI was included as a proxy for cumulative dietary exposure and overall nutritional status. No imputation was performed.

## Statistical analyses

Differences between cases and controls were assessed using two-tailed *t*-tests for continuous variables and Chi-square test for categorical variables. Conditional logistic regression was used to estimate adjusted odds ratios (aORs) for the association between medication use and upper GI malignancy, accounting for the matched design. Models were further adjusted for potential confounders. To address reverse causality and confounding by indication, we included time-binned medication exposures and constructed additional models excluding the year before diagnosis. Diagnostic codes for GI conditions (e.g., gastritis, GERD, peptic ulcer disease) were included to control for symptomatic indications that may precede cancer diagnosis.

Analyses were performed using R version 4.4.0 (R Foundation for Statistical Computing). Data extraction and preprocessing were conducted using structured query language (SQL) and Python version 3.11 scripts developed by the Leumit Research Institute.

A two-sided $P < 0.05$ was considered statistically significant.

### Protocol and analysis plan

This retrospective case–control study did not have a prospectively registered protocol or analysis plan. The initial study design specified: (i) case definition and inclusion criteria; (ii) a 1:10 matching scheme on sex, ethnic sector, socioeconomic status, and year of enrollment with nearest date-of-birth selection among eligible controls; (iii) exposure modeling using discrete pre-diagnosis windows; and (iv) a primary multivariable conditional logistic regression adjusted for prespecified demographic, lifestyle, and clinical covariates.

During peer review, we conducted additional, post hoc analyses to address reviewer comments; these are reported as sensitivity or subgroup analyses and did not change the study's main conclusions. Specifically, we (1) expanded covariate adjustment where available in the EHR (e.g., alcohol use, healthcare utilization, refined SES and BMI categories, pregnancy history, *H. pylori* diagnosis); (2) added a gastric cancer–only subgroup analysis; (3) performed a cumulative-exposure sensitivity analysis by stratifying prescription counts within each time window (median split); and (4) reported post-match balance diagnostics and revised Table 1 testing (N/A *p*-values for matched variables; one global *p*-value for multi-level categorical variables). These changes were made to improve clarity, address potential residual confounding, and enhance transparency.

This study is reported as per the RECORD (Reporting of Studies Conducted Using Observational Routinely-Collected Data) guideline (S1 Checklist).

### Ethical approval

This study was approved by the Leumit Health Services Institutional Review Board (IRB) with a waiver of informed consent (approval number: LEU-0010–21). The waiver was justified on the basis that this large retrospective study used de-identified clinical data analyzed anonymously.

## Results

Fig 1 illustrates the flowchart used to construct the study cohort. The study included 875 patients diagnosed with upper GI cancer and 8,750 matched cancer-free controls (Table 1). Cases and controls were well matched by age, sex, ethnic sector, and year of enrollment. The mean age in both groups was 63.0 years (standard deviation [SD] 11.9), and 37.5% were female. Ethnic sector distribution was matched by design, with 12.2% Arab, 10.3% Ultra-Orthodox Jewish, and 77.5% from the general population.

Individuals with cancer were more likely than controls to be underweight (3.0% versus 1.1%) and less likely to be obese (25.9% versus 31.7%). Current smoking was more common among cases (25.2% versus 19.7%, $p < 0.001$), and the average socioeconomic status score was slightly lower (9.20 versus 9.56, $p = 0.004$). Hypertension was slightly less prevalent in cases (20.3% versus 23.8%, for stage 2 hypertension), and mean hemoglobin (13.2 versus 14.0 g/dL) and high-density lipoprotein (HDL) cholesterol (47.5 versus 49.7 mg/dL) levels were lower among cases, though creatinine, low-density lipoprotein (LDL) cholesterol, and hemoglobin A1c (HbA1c) levels were similar between groups.

Medication exposure and GI diagnoses in the years before the index date are summarized in Table 2. PPI use was significantly more common among cases than controls, including esomeprazole (24.0% versus 8.4%; OR 3.42), omeprazole (59.9% versus 36.3%; OR 2.62), and lansoprazole (12.5% versus 6.8%; OR 1.94). Famotidine use was also more frequent among cases, while calcium carbonate showed no difference. Individuals with cancer were also more likely to

**Table 1. Demographic and clinical characteristics of individuals with upper gastrointestinal cancer and matched controls.**

| | | Individuals with cancer | control | *SMD* | *P* value |
|---|---|---|---|---|---|
| **N** | | 875 | 8,750 | | |
| **Sex** | **Female** | 328 (37.5%) | 3,280 (37.5%) | | N/A matched |
| | **Male** | 547 (62.5%) | 5,470 (62.5%) | | |
| **Age (years)** | | 63.0 ± 11.9 | 63.0 ± 11.9 | 0.000 | 0.999 |
| **Body Mass Index (BMI)** | | 27.0 ± 5.1 | 28.2 ± 5.2 | −0.235 | <0.001 |
| **BMI category (kg/m²)** | **<18.5 Underweight** | 23 (3.03%) | 87 (1.07%) | | <0.001 |
| | **18.5–24.9 Normal** | 258 (34.04%) | 2,181 (26.74%) | | |
| | **25–29.9 Overweight** | 281 (37.07%) | 3,306 (40.53%) | | |
| | **≥30 Obese** | 196 (25.86%) | 2,582 (31.66%) | | |
| **BP systolic (mmHg)** | | 129 ± 18 | 131 ± 21 | −0.105 | 0.003 |
| **BP diastolic (mmHg)** | | 76.4 ± 9.9 | 77.7 ± 9.7 | −0.130 | <0.001 |
| **BP category** | **Hypertension 1** | 204 (23.9%) | 2,135 (24.8%) | | 0.025 |
| | **Hypertension 2** | 173 (20.3%) | 2,043 (23.8%) | | |
| | **Normal BP** | 477 (55.9%) | 4,417 (51.4%) | | |
| **Smoking status** | **Nonsmoker** | 508 (71.0%) | 6,208 (76.8%) | | <0.001 |
| | **Past smoker** | 27 (3.78%) | 280 (3.46%) | | 0.670 |
| | **Smoker** | 180 (25.2%) | 1,595 (19.7%) | | <0.001 |
| **Ethnic sector** | **Arab** | 107 (12.2%) | 1,070 (12.2%) | | N/A matched |
| | **General** | 678 (77.5%) | 6,780 (77.5%) | | |
| | **Jewish Ultra-orthodox** | 90 (10.3%) | 900 (10.3%) | | |
| **Creatinine (mg/dL)** | | 0.91 ± 0.44 | 0.91 ± 0.43 | 0.008 | 0.820 |
| **eGFR MDRD** | **(mL/min/1.73m²)** | 84.1 ± 24.5 | 83.4 ± 24.2 | 0.031 | 0.388 |
| **Triglycerides** | **(mg/dL)** | 129 ± 65 | 134 ± 77 | −0.076 | 0.039 |
| **Hemoglobin A1c** | **(%)** | 6.11 ± 1.07 | 6.16 ± 1.12 | −0.045 | 0.292 |
| **HDL Cholesterol** | **(mg/dL)** | 47.5 ± 13.0 | 49.7 ± 13.8 | −0.159 | <0.001 |
| **LDL Cholesterol** | **(mg/dL)** | 114 ± 36 | 115 ± 36 | −0.021 | 0.576 |
| **Macroalbuminuria** | | 26 (2.97%) | 229 (2.62%) | | 0.534 |
| **Microalbuminuria** | | 112 (12.8%) | 1,100 (12.6%) | | 0.846 |
| **Hemoglobin** | **(g/dL)** | 13.2 ± 2.0 | 14.0 ± 1.5 | −0.507 | <0.001 |

Demographic and clinical characteristics of patients diagnosed with upper gastrointestinal (GI) cancer (cases) and their matched controls. Exact matching was performed on sex, ethnic sector, year of first membership in Leumit Health Services, and year of birth, closest match was taken for the date of birth. Data are shown as counts and percentages or means with standard deviations, as appropriate. Differences between cases and controls were assessed using *t* test for continuous variables and Chi-Square for categorical variables.

SMD: Standardized Mean Difference.

N/A: Not applicable, for variables used to match case and controls.

have been diagnosed with upper GI conditions, including peptic ulcer disease, abdominal pain, gastritis, gastroesophageal reflux disease (GERD), and *H. pylori* infection.

Multivariable conditional logistic regression models adjusting for demographic and clinical factors found that recent PPI use (within 5 years) was associated with increased odds of cancer (Fig 2A). Esomeprazole had the strongest association (adjusted OR 4.01, 95% CI 3.20–5.03, $p < 0.001$), followed by omeprazole (aOR 2.38, 95% CI 1.99–2.85, $p < 0.001$). The association for lansoprazole was borderline and did not reach statistical significance (aOR 1.26, 95% CI 0.96–1.66, $p = 0.096$). In contrast, famotidine and calcium carbonate were not significantly associated with cancer odds.

PLOS Medicine

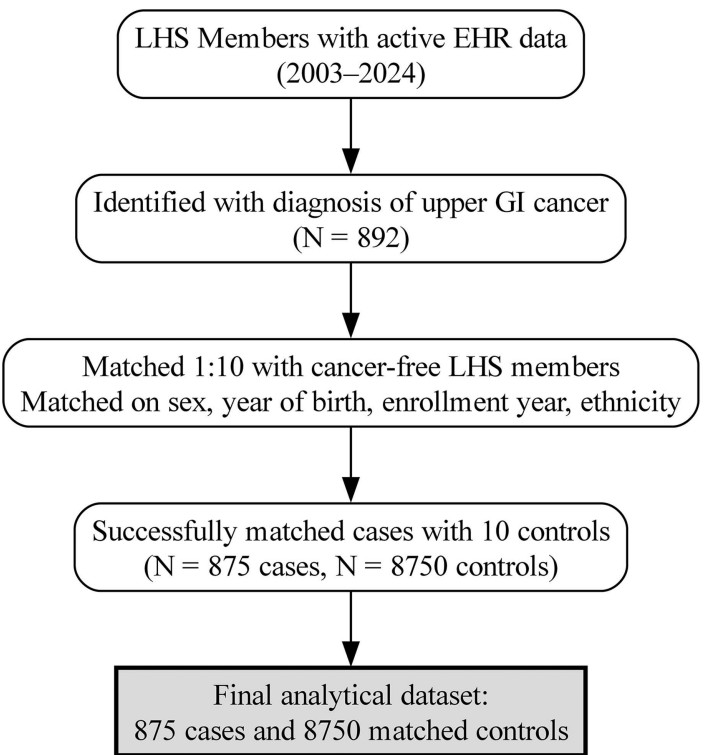

**Fig 1. Flow diagram of the cohort.** Adults (18−80 years) enrolled in Leumit Health Services (LHS) with electronic health records (EHRs) between 2003–2024 were eligible. Incident upper gastrointestinal (GI) cancers (stomach, esophagus, or small intestine) were identified via the national cancer registry and ICD-9–coded EHR diagnoses. Exclusions included prior malignant neoplasm before the index date, <12 months of enrollment, and missing key covariates. The final analytic set comprised 875 cases matched 1:10 to 8,750 cancer-free controls on sex, year of birth, year of enrollment, ethnic sector, and age were closely matched using nearest date of birth. Box totals indicate unique individuals retained at each step. Abbreviations: EHR, electronic health record; GI, gastrointestinal; ICD-9, International Classification of Diseases, Ninth Revision; LHS, Leumit Health Services.

In time-stratified analyses (Fig 2B), the highest odds were observed with PPI use within the year preceding cancer diagnosis, especially in the 0–0.1 year and 0.1–0.5 year windows. For esomeprazole, the adjusted odds ratios were 7.70 (95% CI 5.25–11.31), $p < 0.001$ and 5.80 (95% CI 3.88–8.65), $p < 0.001$, respectively. Risk declined with increasing time since last use, and no significant associations were found for PPI use more than 1 year before diagnosis.

In a model excluding PPI use during the final year and adjusting for GI diagnoses (Fig 3), no positive association remained between any PPI and cancer odds (e.g., omeprazole 1–2 years before aOR=1.17, 95% CI 0.92–1.49, $p = 0.19$). Instead, exposure to omeprazole and lansoprazole in the 3–10 year window was associated with reduced odds of cancer (omeprazole: aOR 0.62, 95% CI 0.50–0.77, $p < 0.001$; lansoprazole: aOR 0.64, 95% CI 0.45–0.91, $p = 0.013$). Esomeprazole showed a similar though nonsignificant tendency toward reduced odds (aOR 0.72, 95% CI 0.48–1.09, $p = 0.12$). Diagnoses such as peptic ulcer disease (aOR 3.74, 95% CI 2.69–5.21, $p < 0.001$), abdominal pain (aOR 2.90, 95% CI 2.37–3.55, $p < 0.001$), alcohol use (aOR 2.35, 95% CI 1.17–4.70, $p = 0.02$), *H. pylori* (1.43, 95% CI 1.13–1.80, p = 0.003), GERD (aOR 1.66, 95% CI 1.32–2.09, $p < 0.001$), and gastritis (aOR 1.65, 95% CI 1.29–2.11, $p < 0.001$) remained strongly associated with cancer.

To evaluate potential dose-response relationships, we conducted a sensitivity analysis stratifying PPI exposure within each time window based on cumulative prescription counts, dichotomized at the median observed in the study cohort. This approach allowed us to distinguish between higher and lower cumulative users while maintaining consistent

Table 2. Comparative analysis of medication exposure and gastrointestinal diagnoses recorded in the 5 years before the index date among individuals with upper gastrointestinal cancer and matched controls.

| | Individuals with cancer (%) | Control (%) | Odds Ratio | 95% Confidence Interval | P Value |
|---|---|---|---|---|---|
| **Medication purchase in the last 5 years** | | | | | |
| Famotidine PO | 139 (15.9%) | 1,072 (12.3%) | 1.35 | [1.11 to 1.64] | 0.0027 |
| Omeprazole PO | 524 (59.9%) | 3,175 (36.3%) | 2.62 | [2.27 to 3.03] | <0.0001 |
| Lansoprazole PO | 109 (12.5%) | 598 (6.8%) | 1.94 | [1.55 to 2.42] | <0.0001 |
| Esomeprazole PO | 210 (24.0%) | 739 (8.4%) | 3.42 | [2.87 to 4.08] | <0.0001 |
| Calcium Carbonate PO | 24 (2.7%) | 201 (2.3%) | 1.20 | [0.75 to 1.85] | 0.4104 |
| **Diagnosis recorded** | | | | | |
| Peptic Ulcer Disease | 90 (10.3%) | 201 (2.3%) | 4.87 | [3.72 to 6.35] | <0.0001 |
| Abdominal pain | 607 (69.4%) | 3,961 (45.3%) | 2.74 | [2.35 to 3.19] | <0.0001 |
| Alcohol use | 17 (1.94%) | 56 (0.64%) | 3.08 | [1.67 to 5.40] | <0.0001 |
| GERD | 186 (21.3%) | 1,159 (13.2%) | 1.77 | [1.48 to 2.11] | <0.0001 |
| Gastritis | 181 (20.7%) | 864 (9.9%) | 2.38 | [1.98 to 2.85] | <0.0001 |
| *H. Pylori* | 187 (21.4%) | 954 (10.9%) | 2.22 | [1.85 to 2.65] | <0.0001 |

*P*-values calculated using Fisher's exact test. Medication exposure reflects at least one pharmacy purchase in the 5 years preceding the index date. Diagnoses include any recorded occurrence in a 10-year window.

time-based exposure definitions. The analysis revealed the same temporal patterns as our main findings. Notably, for remote PPI use (>3 years before diagnosis), higher cumulative exposure was associated with a more pronounced protective association. Although statistical significance was attenuated due to halved sample sizes, the direction and magnitude of effects were consistent. A forest plot is presented in S1 Fig.

To specifically assess associations with gastric cancer, we conducted a subgroup analysis restricted to 701 cases of gastric cancer and their 7,010 matched controls. Results were consistent with the primary analysis: after excluding the year prior to diagnosis and adjusting for symptom-related diagnoses, we found no evidence of increased cancer odds associated with recent or long-term PPI use. For remote PPI use (3–10 years before diagnosis), lansoprazole and omeprazole showed a similar tendency toward reduced odds (S2 Fig).

## Discussion

In this large matched case-control study based on EHRs from a large health organization, we evaluated the association between PPI use and the odds of upper gastrointestinal malignancies. In models without adjustment for symptom-related diagnoses, PPI use was associated with higher odds of cancer. In time window analyses, use in the 0–6 months preceding diagnosis was associated with significantly elevated odds; however, once recent exposure was accounted for, remote use beyond this period was not associated with increased odds. After excluding the final year before diagnosis and adjusting for symptom-related diagnoses, earlier use (>1 year before diagnosis) showed no evidence of increased odds, and remote use (>3 years before diagnosis) was associated with lower odds, particularly for omeprazole and lansoprazole. The observation that elevated odds appeared only in unadjusted analyses, but disappeared after accounting for diagnostic context and timing supports the interpretation that harmful signals in studies that omit symptom adjustment or fail to model exposure with discrete time windows likely reflect confounding by indication and protopathic bias rather than a causal effect.

Reverse causation, also known as protopathic bias, is a form of bias that arises when a medication is prescribed for early symptoms of a disease that has not yet been diagnosed. In this context, early manifestations of a process associated with cancer risk (such as dyspepsia, reflux, or abdominal discomfort) may lead to the initiation of acid-suppressive

## A Medication exposure in the 5 years preceding cancer diagnosis

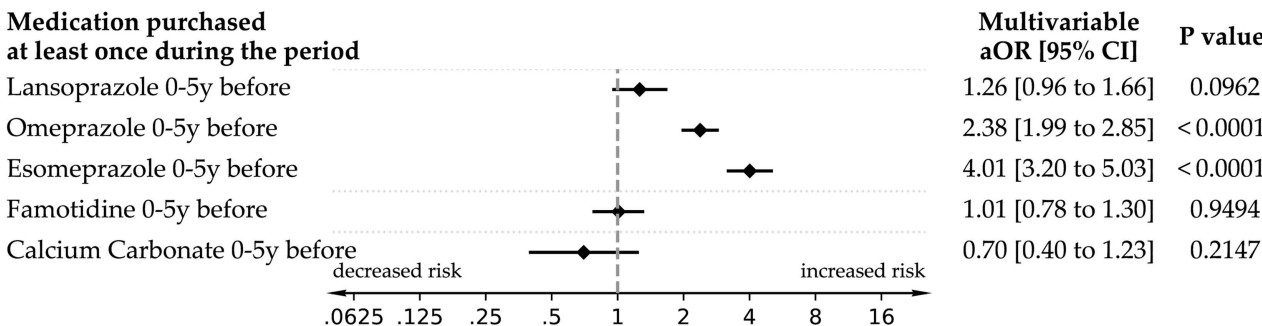

| Medication purchased at least once during the period | Multivariable aOR [95% CI] | P value |
|---|---|---|
| Lansoprazole 0-5y before | 1.26 [0.96 to 1.66] | 0.0962 |
| Omeprazole 0-5y before | 2.38 [1.99 to 2.85] | < 0.0001 |
| Esomeprazole 0-5y before | 4.01 [3.20 to 5.03] | < 0.0001 |
| Famotidine 0-5y before | 1.01 [0.78 to 1.30] | 0.9494 |
| Calcium Carbonate 0-5y before | 0.70 [0.40 to 1.23] | 0.2147 |

## B Time-resolved medication exposure by interval before cancer diagnosis

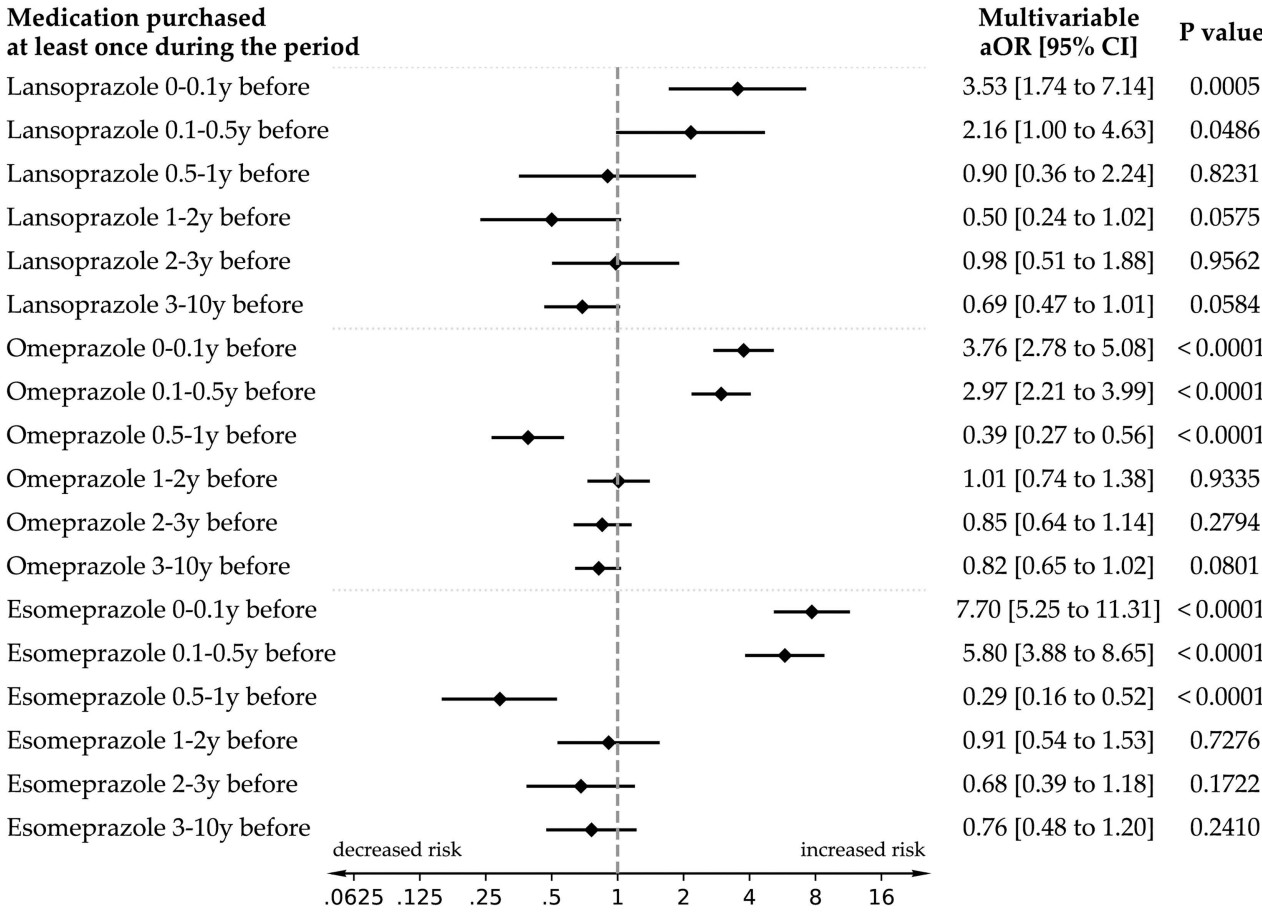

| Medication purchased at least once during the period | Multivariable aOR [95% CI] | P value |
|---|---|---|
| Lansoprazole 0-0.1y before | 3.53 [1.74 to 7.14] | 0.0005 |
| Lansoprazole 0.1-0.5y before | 2.16 [1.00 to 4.63] | 0.0486 |
| Lansoprazole 0.5-1y before | 0.90 [0.36 to 2.24] | 0.8231 |
| Lansoprazole 1-2y before | 0.50 [0.24 to 1.02] | 0.0575 |
| Lansoprazole 2-3y before | 0.98 [0.51 to 1.88] | 0.9562 |
| Lansoprazole 3-10y before | 0.69 [0.47 to 1.01] | 0.0584 |
| Omeprazole 0-0.1y before | 3.76 [2.78 to 5.08] | < 0.0001 |
| Omeprazole 0.1-0.5y before | 2.97 [2.21 to 3.99] | < 0.0001 |
| Omeprazole 0.5-1y before | 0.39 [0.27 to 0.56] | < 0.0001 |
| Omeprazole 1-2y before | 1.01 [0.74 to 1.38] | 0.9335 |
| Omeprazole 2-3y before | 0.85 [0.64 to 1.14] | 0.2794 |
| Omeprazole 3-10y before | 0.82 [0.65 to 1.02] | 0.0801 |
| Esomeprazole 0-0.1y before | 7.70 [5.25 to 11.31] | < 0.0001 |
| Esomeprazole 0.1-0.5y before | 5.80 [3.88 to 8.65] | < 0.0001 |
| Esomeprazole 0.5-1y before | 0.29 [0.16 to 0.52] | < 0.0001 |
| Esomeprazole 1-2y before | 0.91 [0.54 to 1.53] | 0.7276 |
| Esomeprazole 2-3y before | 0.68 [0.39 to 1.18] | 0.1722 |
| Esomeprazole 3-10y before | 0.76 [0.48 to 1.20] | 0.2410 |

**Fig 2. Association between PPI use and upper gastrointestinal cancer: impact of exposure timing.** Forest plots illustrating the association between proton pump inhibitor (PPI) and related medication use and the risk of upper gastrointestinal (GI) cancer, based on multivariable conditional logistic regression models adjusted for age, smoking, Body Mass Index (BMI) category, socioeconomic status, healthcare utilization, and pregnancy history. Panel A shows odds ratios (ORs) for cancer associated with any medication exposure during the years preceding diagnosis, adjusted for demographic and clinical covariates. Panel B presents time-resolved associations across multiple exposure windows, highlighting elevated risk during the final year before diagnosis and reduced or null risk for more distant exposures after adjustment for recent exposure.

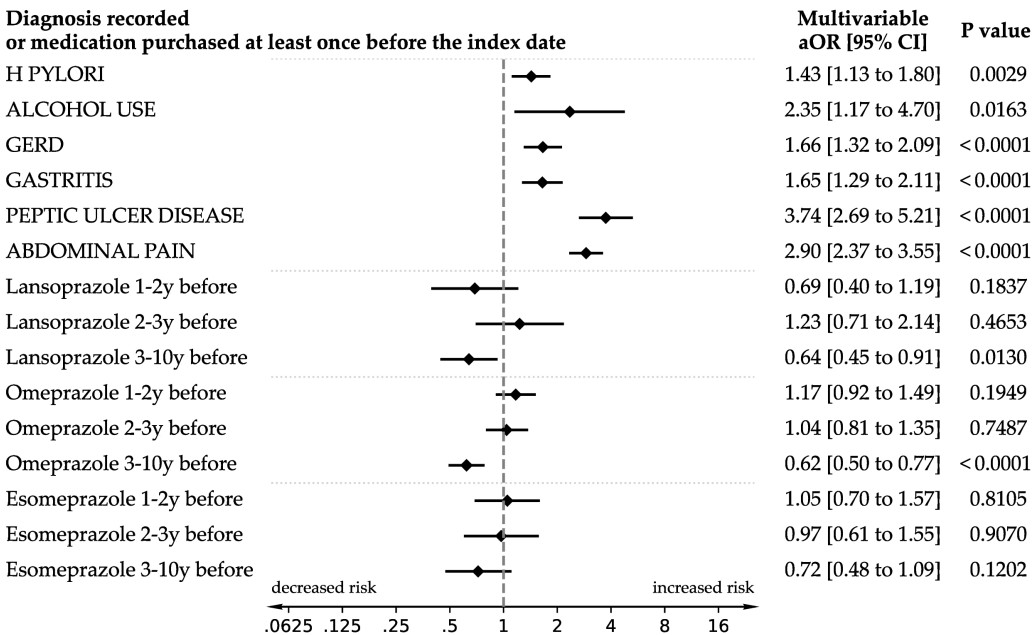

**Fig 3. Association between past PPI use and upper gastrointestinal cancer risk after adjustment for gastrointestinal diagnoses.** Forest plot showing adjusted odds ratios (aORs) and 95% confidence intervals for the association between proton pump inhibitor (PPI) use at different time windows prior to 1 year before the index date (1–2 years, 2–3 years, and 3–10 years) and the risk of upper gastrointestinal (GI) cancer. Models are adjusted for age, smoking, BMI category, socioeconomic status, healthcare utilization, pregnancy history (women), *H. pylori* diagnosis, alcohol use, and upper GI symptom-related diagnoses, including abdominal pain and GERD (gastroesophageal reflux disease). Time windows are treated as separate binary exposures, with medication exposure defined by ≥1 pharmacy purchase during the interval. The analysis uses multivariable conditional logistic regression within a matched case-control framework (1:10 matching). Findings highlight that after adjusting for underlying GI conditions, PPI use more than 1 year prior to cancer diagnosis is not associated with increased cancer risk—and in several cases is associated with decreased risk.

therapy, creating a spurious association between the medication and subsequent cancer diagnosis. This can create a misleading association between the medication and the eventual cancer diagnosis. In our dataset, the adjusted odds of cancer were highest for PPI use in the 0–0.1 year and 0.1–0.5 year intervals before diagnosis, consistent with this mechanism. Many previous observational studies that reported increased cancer odds with PPI use did not account for timing of exposure [12–19] or for preexisting GI conditions that prompted treatment [1,32]. Consequently, they may have conflated symptomatic treatment with causative risk.

Our study addressed these limitations in several ways. First, we used a high-resolution, time-anchored dataset that allowed for temporal disaggregation of medication exposure windows. This made it possible to identify that only recent PPI use was associated with increased odds, while more remote exposure had no such association—or even appeared protective after accounting for recent exposure. Second, we adjusted for a comprehensive set of GI diagnoses frequently preceding PPI use, including *H. pylori* infection, peptic ulcer disease, gastritis, GERD, alcohol use, abdominal pain. When these clinical indications were included in the model and recent medication exposure was excluded, the association between PPI use and cancer disappeared or reversed, with PPIs (omeprazole and lansoprazole) showing a statistically significant protective association.

These findings also align with biological plausibility. Chronic mucosal injury—such as that caused by persistent acid reflux or untreated *H. pylori* infection—is a well-established risk factor for gastric and esophageal adenocarcinoma [36]. PPIs reduce gastric acid secretion, promote mucosal healing, and are effective in mitigating these precancerous processes [37]. Prior studies have shown that PPI therapy in patients with Barrett's esophagus or chronic gastritis may

reduce progression to dysplasia or malignancy [38,39]. The observed protective association with long-term PPI use in our study could reflect effective acid suppression and mucosal stabilization in patients at elevated baseline risk.

An additional consideration is the interaction between PPI therapy and *H. pylori* infection. PPIs are an integral part of *H. pylori* eradication regimens, which also include antibiotic therapy. Successful eradication [40], followed by mucosal healing under acid suppression, reduces the risk of atrophic gastritis and intestinal metaplasia. However, when *H. pylori* persists—particularly under prolonged acid suppression—gastrin levels rise, and altered gastric pH may allow the pathogen to persist in an inflammatory niche [23]. Thus, the net effect of long-term PPI use may vary depending on whether *H. pylori* has been eradicated or remains untreated. Our data could not directly assess *H. pylori* treatment success, but the protective associations seen with remote PPI use may reflect earlier eradication or stabilization of chronic inflammation.

Persistent hypergastrinemia has been implicated as a trophic stimulus for gastric mucosal proliferation in experimental models [22]. Rare hereditary cases carrying germline ATP4A mutations develop gastric neuroendocrine tumors and adenocarcinoma at young age [41]. However, these conditions represent extreme pathophysiology distinct from routine PPI exposure. Moreover, long-term PPI users in clinical practice undergo more frequent endoscopic evaluation, which could lead to earlier tumor detection (surveillance bias). The lack of excess cancer among such users in our data therefore strengthens the inference that PPIs themselves are unlikely to be causative.

This study has several strengths, including a large sample size; comprehensive electronic health records spanning over two decades that capture pharmacy purchases and ICD-coded diagnoses at each clinical encounter; precise malignancy onset dating that enabled rigorously timed case-control matching; high-resolution temporal analysis of medication exposure; and robust adjustment for potential confounders, including age, sex, smoking status, alcohol consumption, BMI, socioeconomic status (SES), healthcare utilization, and pregnancy history. Although we lacked direct measures of dietary preferences or family-history, BMI serves as a validated proxy for long-term nutritional status and energy intake in Israeli and other populations [42,43]. Detailed dietary preferences were not recorded in the EHR; however, body mass index and socioeconomic indicators remain the most feasible proxies for cumulative nutritional exposure in this setting. We have therefore made the best possible attempt, within the constraints of our health system, to mitigate dietary confounding. These measures may even provide greater objectivity than dietary questionnaires used in other studies, which are prone to recall and reporting bias. Nevertheless, residual confounding from unmeasured or imperfectly captured factors, such as specific dietary exposures or adherence to *H. pylori* eradication protocols, remains a potential limitation. While ICD-coded diagnoses were assigned by treating physicians at the time of encounter, minimizing misclassification and recall bias, some clinically relevant factors, such as symptom severity or undocumented conditions, may not be fully captured within structured EHR data fields. Additionally, pharmacy purchase data may not fully capture actual medication adherence, which could introduce nondifferential exposure misclassification. Also, our study is based on a 10-year observation window, reflecting the available longitudinal EHR span and may be insufficient to capture very long-latency pathways linking childhood *H. pylori* to later-life malignancy [41]. Findings should therefore be interpreted in the context of this time horizon. Our database also does not include genotyping information, and thus, we were unable to perform Mendelian randomization analyses to explore causal associations using genetic instrumental variables.

Although our cohort was ethnically, culturally, and socioeconomically diverse, regional differences in diet, infection prevalence, and healthcare access could still influence upper gastrointestinal cancer incidence. The generalizability of these findings to populations with different risk profiles—such as those in East Asia, where gastric cancer incidence is higher—remains uncertain. Given the cohort's age distribution, the results primarily inform older adults and should be extrapolated to younger populations with caution. Replication in other geographic and ethnic settings, including Europe, North America, and Asia, is warranted to confirm external validity.

In conclusion, this observational study demonstrates that the apparent association between long-term PPI use and upper gastrointestinal cancer can be explained by confounding by indication and reverse causation, rather than by a true pharmacologic effect. After adjustment for diagnostic context and timing, we did not detect a statistically significant association

between PPI use and increased cancer odds. Some analyses of remote use (3–10 years before diagnosis) yielded adjusted odds ratios below 1.0, notably for omeprazole and lansoprazole, suggesting a possible protective association.

## Supporting information

**S1 Checklist. RECORD checklist (completed).** Checklist documenting adherence to the RECORD reporting guideline for studies using routinely collected health data, corresponding to the manuscript. The checklist lists each item and the location in the manuscript; no additional data are included.
(DOCX)

**S1 Fig. Subgroup analysis: gastric cancer only.** Forest plot of adjusted odds ratios (aORs) for PPI use and registry-confirmed gastric cancer (701 cases; 7,010 matched controls), by exposure windows (1–2 years, 2–3 years, 3–10 years before diagnosis). Estimates from conditional logistic regression within 1:10 matched sets; models adjust for age, smoking, alcohol use, BMI category, socioeconomic status, healthcare utilization, pregnancy history (women), *H. pylori* diagnosis, and upper GI symptom-related diagnoses. Error bars indicate 95% confidence intervals (CIs).
(JPG)

**S2 Fig. Dose-response (dichotomized) analysis of PPI purchases.** Forest plot comparing higher versus lower cumulative prescription counts within each exposure window (median split) for each PPI. Models and adjustments as in S1 Fig.; conditional logistic regression within matched sets. Error bars show 95% CIs.
(JPG)

**S1 Table. Medication exposure by time window and GI symptom diagnoses.** Counts/percentages of PPI purchases in each pre-diagnosis window (0–1 month; 2–6 months, 6–12 months, 1–2 years, 2–3 years, 3–10 years) and prevalence of upper GI symptom-related diagnoses recorded before the index date in cases and controls.
(XLSX)

**S2 Table. Tumor anatomic site (registry-documented cases).** Distribution of primary sites among upper GI cancers with a confirmed site in the national cancer registry (counts and percentages).
(XLSX)

**S3 Table. Tumor morphology (registry-documented cases).** Distribution of histologic morphologies for registry-documented upper GI cancers.
(XLSX)

**S4 Table. Tumor differentiation (registry-documented cases).** Distribution of tumor differentiation grades for registry-documented upper GI cancers.
(XLSX)

**S5 Table. ICD-9 diagnosis codes used for upper GI cancers are documented only in the LHS EHR.** List of ICD-9 codes and code groupings used to identify incident upper GI cancer in the electronic health record when no corresponding registry record was available.
(XLSX)

**S6 Table. ICD-9 codes for GI symptoms and related diagnoses.** Code lists for dyspepsia, gastroesophageal reflux disease, abdominal pain, gastritis, peptic ulcer disease, and *H. pylori* infection were used to construct symptom-related covariates. Abbreviations: aOR, adjusted odds ratio; BMI, body mass index; CI, confidence interval; EHR, electronic health record; GI, gastrointestinal; LHS, Leumit Health Services; PPI, proton pump inhibitor.
(XLSX)

## Acknowledgments

The content of this publication does not necessarily reflect the views or policies of the Department of Health and Human Services, nor does mention of trade names, commercial products, or organizations imply endorsement by the US government.

## Author contributions

**Conceptualization:** Ariel Israel.

**Formal analysis:** Ariel Israel.

**Investigation:** Ibrahim O. Sawaid, Zohar Din, Efrat Golan, Eytan Ruppin, Avivit Golan-Cohen, Ilan Green, Eugene Merzon, Shlomo Vinker, Abraham O. Samson, Ariel Israel.

**Methodology:** Ibrahim O. Sawaid, Ariel Israel.

**Supervision:** Abraham O. Samson.

**Visualization:** Ibrahim O. Sawaid.

**Writing – original draft:** Ibrahim O. Sawaid, Zohar Din, Eugene Merzon, Abraham O. Samson, Ariel Israel.

**Writing – review & editing:** Zohar Din, Efrat Golan, Eytan Ruppin, Avivit Golan-Cohen, Ilan Green, Eugene Merzon, Shlomo Vinker, Abraham O. Samson, Ariel Israel.

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
