## [Editor Report · Decision Letter 0]

4 Aug 2025

Dear Dr Israel, 

Thank you for submitting your manuscript entitled "Association between Proton Pump Inhibitor Use and Upper Gastrointestinal Cancer: A Matched Case-Control Study Accounting for Reverse Causation and Confounding by Indication" for consideration by PLOS Medicine.

Your manuscript has now been evaluated by the PLOS Medicine editorial staff as well as by an academic editor with relevant expertise and I am writing to let you know that we would like to send your submission out for external peer review.

For clinical studies, please upload a copy of your trial study protocol as a supporting information file. The study protocol should be the version submitted for approval to the institutional review board or ethics committee, should include any amendments to the study protocol, as well as the date of their approval by the institutional review or ethics committee. Please also detail any deviations from the study protocol in the Methods section of your manuscript. The editors will consider the protocol and study conduct prior to a final decision for external review. 

Please re-submit your manuscript within two working days, i.e. by Aug 06 2025 11:59PM.

Kind regards,

Jennifer Thorley

PLOS Medicine

---

## [Decision Letter · Decision Letter 1]

9 Sep 2025

Dear Dr Israel,

Many thanks for submitting your manuscript "Association between Proton Pump Inhibitor Use and Upper Gastrointestinal Cancer: A Matched Case-Control Study Accounting for Reverse Causation and Confounding by Indication" (PMEDICINE-D-25-02605R1) to PLOS Medicine. The paper has been reviewed by subject experts and a statistician; their comments are included below and can also be accessed here: https://www.editorialmanager.com/pmedicine/l.asp?i=1030064&l=J0J8Y8OI

As you will see, the reviewers find that the study covers an important topic while also noting major methodological issues. After discussing the paper with the editorial team and an academic editor with relevant expertise, I'm pleased to invite you to revise the paper in response to the reviewers' comments. We plan to send the revised paper to some or all of the original reviewers, and we cannot provide any guarantees at this stage regarding publication.

We ask that you submit your revision by Sep 30 2025. However, if this deadline is not feasible, please contact me by email, and we can discuss a suitable alternative.

Don't hesitate to contact me directly with any questions (atosun@plos.org). 

Best regards, 

Alexandra 

Alexandra Tosun, PhD 

Senior Editor

PLOS Medicine

atosun@plos.org

Comments from the academic editor:

This is a timely and important topic, as a wide body of research has implicated PPIs as potential risk factors for a variety of conditions. In this well-designed and carefully analyzed study, the authors demonstrate that the initially observed risk was explained by underlying methodological biases, including protopathic bias. The findings are clinically relevant and advance the field methodologically.

The reviewer comments are insightful and constructive. I agree with the recommendations, particularly those from the first reviewer who highlighted important considerations for strengthening the analysis. I also agree that generalizability to other populations, such as Asian countries with a higher incidence of gastric cancer, is a limitation. This could be addressed more explicitly in the discussion.

Comments from the reviewers: 

Reviewer #1: Review on: Association between Proton Pump Inhibitor Use and Upper Gastrointestinal Cancer: A Matched Case-Control Study Accounting for Reverse Causation and Confounding by Indication

This study leveraged national Israeli electronic health records in a matched case-control design to evaluate the association between PPI use and upper gastrointestinal (GI) cancer, explicitly addressing confounding by indication and reverse causation. Initial analyses suggested increased risk with Proton Pump Inhibitor (PPI) use in the five years before diagnosis, but the signal was concentrated in the six months preceding diagnosis; after excluding the final year and adjusting for symptom-related diagnoses, the association disappeared. More remote PPI exposure was associated with lower risk. Overall, the findings indicate that previously reported harmful associations are largely driven by potential bias.

The authors address a timely and controversial topic regarding PPI use. By stratifying exposure across different time windows, they aim to demonstrate that the observed association between PPI use and upper GI cancers is primarily driven by confounding by indication and reverse causation. While this overarching conclusion is plausible, the current analyses do not yet provide sufficiently robust support. The manuscript would be strengthened by the following:

1.Insufficient covariate adjustment. Beyond the variables included, several established risk factors for upper GI cancers were not accounted for, such as dietary patterns, family history, alcohol consumption, and overall comorbidity burden. Please consider incorporating these variables where possible and conducting sensitivity analyses to mitigate residual confounding. 

2.Missing dose-response assessment. Exposure was defined as >= 1 pharmacy purchase within a given time window, which may conflate occasional and long-term users. Consider incorporating measures of cumulative exposure (e.g., total prescription days). Prior studies commonly examine dose metrics in relation to upper GI cancer incidence. To more comprehensively argue against a harmful association, please include dose-response analyses, including tests for threshold effects and potential nonlinearity. 

3.Insufficient outcome stratification. The Introduction emphasizes prior work on PPIs and gastric cancer. To align with this framing and enhance specificity, please stratify outcomes by anatomic site and report site-specific estimates. 

4.Limited description of the matching strategy. Please justify the choice of the 1:10 matching ratio and report post-match balance diagnostics and corresponding statistical handling. 

5.Abbreviation consistency. Provide full terms at first mention only and use standardized abbreviations consistently thereafter. 

6.Study design extensions. To further address the possibility of a causal effect of long-term PPI use, consider Mendelian randomization analyses, using genetic variants as instrumental variables where feasible. 

7.External validity. The study population is limited to Israel, whereas prior literature is largely from Europe, North America, and Asia. To improve generalizability, consider validating findings in additional regions; if not feasible, expand the discussion of this limitation.

Overall, while the research topic is timely and valuable, the current analysis is insufficient and does not meet the PLOS Medicine's publication standards. For this reason, I recommend that this manuscript not be accepted for publication. I hope these comments will help the author team improve the quality of the manuscript.

Reviewer #2: In general, this is a well-conducted retrospective study that aims to address whether previously reported associations between receipt of PPIs and subsequent GI cancer might be due to reverse causation. Two strengths are the data source and the careful consideration of straightforward, but appropriate statistical analyses. My comments are only minor.

First, it wasn't exactly clear what variables were used for the matching. For instance, the matching section (page 8) implies that socioeconomic status was used for matching, but the figures/tables implied that socioeconomic status was not used for matching, and that furthermore, age /was/ used for matching. Be clear which matching variables were used; for fuzzy or imperfect matches, also provide some quantification regarding the degree to which the controls may have differed from the cases.

Regarding Table 1, for any variables that were matched on, by design they will have the same distribution between the cases and controls; it may be worth simply having N/A or - or some such in the p-value column since it is not worth describing potential differences between them. Furthermore, for the categorical variables with multiple categories, I found it very strange that a Fisher's Exact Test was used for each of the various levels instead of the entire variable at once (e.g., a single p-value looking at associations between BMI category and case/control status, instead of the four group-specific p-values). You are simply looking for an association between a four-level categorical variable (BMI category) and a binary variable (case/control). There should only be a single p-value.

For the multivariable models, I didn't find explicitly what the non-exposure variables that were controlled for were. Are they the variables listed in the "Covariates" subsection of the methods? If so, I found it a bit odd that you did not adjust for some variables which were found to have been statistically significantly different between the cases and controls (e.g., BMI category). Please be explicit regarding which variables were in each of your multivariable logistic regression models, and consider including a more comprehensive set of variables to control for.

Finally, there is a bit of a misstatement in the conclusion: "When appropriately adjusted for diagnostic context and timing, PPI use is not associated with increased cancer risk and may be associated with lower risk when used in the more distant past." implies that there is no association. However, absence of evidence is not evidence of absence; it is more appropriate to conclude that you failed to find an association between these two variables after adjusting for diagnostic context and timing, not that affirmatively no such association exists.

Reviewer #3: The authors underestimates the previous concerns regarding use of PPI regarding gastric cancer by only referring to one paper on PPI use in older adults. In the discussion it should be referred to the long-term studies in the rats in the middle of the eighties describing neuroendocrine tumors. Similar finding with the insurmountable histamine-2 blocker loxtidine so it was evident that the tumors were due to profound acid inhibition leading to the trophic effect of gastrin on its target cell, the ECL cell. Later it has been shown that every condition with long-term hypergastrinemia in man and animals predisposes to gastric cancer. Even the carcinogenic effect of H. pylori infection may be due to hypergastrinemia secondary to hypergastrinemia (Scand J Gastroenterol. 2025 Jul;60(7):652-663. doi: 10.1080/00365521.2025.2509094. Epub 2025 May 24.PMID: 40411354 ). H. pylori infection most often in childhood, and gastric cancer is mainly a disease of old age. Accordingly, the observation period of 10 years as in the present study is far too short. Also, there is little reason to include the esophagus and gut in studies related to PPI carcinogenesis (for the gut there is atheoretical possibility that microbiological changes due to PPI could predispose to cancer). The relatively short latency of gastric cancer by using profound acid inhibitors after H. pylori eradication indicates additive effect via the same mechanism ( hypergastrinemia) (Cheung et al. Gut 2018 ; 67 : 28-35). Hypergastrinemia alone induces malignant gastric neoplasia after at least two decades (Calvete et al. Hum Mol Genet 2015 24:2914-22) Furthermore, the H. pylori positivity was relatively low among controls as well as cancer patients. What method was used? The present study confirms that initiating PPIs in relatively old people not H. pylori infected for some years is relatively safe with respect to gastric cancer. The results must not be transformed to the same conclusion in younger individuals.

Any attachments provided with reviews can be seen via the following link: https://www.editorialmanager.com/pmedicine/l.asp?i=1030064&l=J0J8Y8OI

---

* Please upload any figures associated with your paper as individual TIF or EPS files with 300dpi resolution at resubmission; please read our figure guidelines for more information on our requirements: http://journals.plos.org/plosmedicine/s/figures. While revising your submission, we strongly recommend that you use PLOS's NAAS tool (https://ngplosjournals.pagemajik.ai/artanalysis) to test your figure files. NAAS can convert your figure files to the TIFF file type and meet basic requirements (such as print size, resolution), or provide you with a report on issues that do not meet our requirements and that NAAS cannot fix.

After uploading your figures to PLOS's NAAS tool - https://ngplosjournals.pagemajik.ai/artanalysis, NAAS will process the files provided and display the results in the "Uploaded Files" section of the page as the processing is complete.

If the uploaded figures meet our requirements (or NAAS is able to fix the files to meet our requirements), the figure will be marked as "fixed" above. If NAAS is unable to fix the files, a red "failed" label will appear above.

When NAAS has confirmed that the figure files meet our requirements, please download the file via the download option, and include these NAAS processed figure files when submitting your revised manuscript.

* The Data Availability Statement (DAS) requires revision: If the data are not freely available, please describe briefly the ethical, legal, or contractual restriction that prevents you from sharing it. Please also include an appropriate contact (web or email address) for inquiries (this cannot be a study author).

* Please include the statement on ethical approval in the Methods section of the main text.

FIGURES AND TABLES

SUPPLEMENTARY MATERIAL

REFERENCES

STUDY TYPE-SPECIFIC REQUESTS

* Abstract: Please include the study design, population and setting, number of participants, years during which the study took place (enrollment and follow up), length of follow up, and main outcome measures.

* Please ensure that the study is reported according to the STROBE (or appropriate STOBE extension) guideline (available from: https://www.equator-network.org/reporting-guidelines/strobe) and include the completed STROBE (or STROBE extension) checklist as Supporting Information. Please add the following statement, or similar, to the Methods: "This study is reported as per the Strengthening the Reporting of Observational Studies in Epidemiology (STROBE) guideline (S1 Checklist)." When completing the checklist, please use section and paragraph numbers, rather than page numbers. 

* For all observational studies, in the manuscript text, please indicate: (1) the specific hypotheses you intended to test, (2) the analytical methods by which you planned to test them, (3) the analyses you actually performed, and (4) when reported analyses differ from those that were planned, transparent explanations for differences that affect the reliability of the study's results. If a reported analysis was performed based on an interesting but unanticipated pattern in the data, please be clear that the analysis was data driven. 

* Please state in the Methods section whether the study had a prospective protocol or analysis plan. If a prospective analysis plan (from your funding proposal, IRB or other ethics committee submission, study protocol, or other planning document written before analyzing the data) was used in designing the study, please include the relevant document(s) with your revised manuscript as a Supporting Information file to be published alongside your study and cite it in the Methods section. A legend for this file should be included at the end of your manuscript. If no such document exists, please make sure that the Methods section transparently describes when analyses were planned, and when/why any data-driven changes to analyses took place. Changes in the analysis, including those made in response to peer review comments, should be identified as such in the Methods section of the paper, with rationale.

---

## [Decision Letter · Decision Letter 2]

29 Oct 2025

Dear Dr. Israel,

Thank you very much for re-submitting your manuscript "Association between Proton Pump Inhibitor Use and Upper Gastrointestinal Cancer: A Matched Case-Control Study Accounting for Reverse Causation and Confounding by Indication" (PMEDICINE-D-25-02605R2) for review by PLOS Medicine.

Thank you for your detailed response to the reviewers' and editors’ comments. I have discussed the paper with my colleagues and the academic editor, and it has also been seen again by all three original reviewers. The changes made to the paper were mostly satisfactory to the reviewers. As such, we intend to accept the paper for publication, pending your attention to the reviewers' and editors' comments below in a further revision. When submitting your revised paper, please once again include a detailed point-by-point response to the reviewers’ and editorial comments. The remaining issues that need to be addressed are listed at the end of this email.

In revising the manuscript for further consideration here, please ensure you address the specific points made by each reviewer and the editors. In your rebuttal letter you should indicate your response to the reviewers' and editors' comments and the changes you have made in the manuscript. Please submit a clean version of the paper as the main article file. A version with changes marked must also be uploaded as a marked up manuscript file. Please also check the guidelines for revised papers at http://journals.plos.org/plosmedicine/s/revising-your-manuscript for any that apply to your paper.

We ask that you submit your revision within 1 week (Nov 05 2025). However, if this deadline is not feasible, please contact me by email, and we can discuss a suitable alternative.

Please do not hesitate to contact me directly with any questions (atosun@plos.org).

We look forward to receiving the revised manuscript.  

Sincerely,

Alexandra Tosun, PhD

Senior Editor 

PLOS Medicine

plosmedicine.org

Comments from Reviewers:

Reviewer #1: Thank you for submitting your revised manuscript. The author team has addressed my comments and improved the manuscript. The clarity and completeness of the research have been improved, particularly in the Analysis and Discussion sections. However, there are still some areas where your manuscript requires further improvement. I would like to highlight the following points for the editors and author team's consideration:

Minor comment

1) The inclusion of dietary data should not only focus on nutrient intake but also on individual dietary preferences. Previous studies have shown that certain dietary habits or preferences are associated with an increased risk of diseases such as gastric cancer, risks that cannot be fully reflected by BMI alone. The revised version addresses the lack of dietary data in the limitations section, but the subsequent explanation seems unconvincing.

2) This study relies on ICD-9 codes for disease diagnoses. Given that ICD-10 codes are more widely used in current clinical practice, it would be helpful to clarify whether the database used lacks ICD-10 codes. If the dataset only includes ICD-9 codes, there is a risk of missing certain diagnoses, which could affect the results. I suggest that this point be addressed in the limitations section.

3) While I understand that you acknowledge the limitation of not including data from other regions, I believe the results should still be interpreted with caution. The lack of broader geographic representation may affect the generalizability of your findings, and I suggest that this point be emphasized in the discussion of the implications of the results. Additionally, it would be beneficial to include a statement about the potential for regional differences in disease patterns and outcomes.

4) Previous case-control and cohort studies on PPIs and gastric cancer have shown no statistically significant trend between PPI use and increased gastric cancer risk. It is recommended to add a summary or discussion of relevant articles in the introduction and discussion sections to further clarify the novelty and advantages of the study.

While I sincerely appreciate the authors' efforts in revising the manuscript, I still have some reservations regarding its content. However, in light of the positive feedback from the editors, I have provided additional suggestions for improvement. If the authors address these concerns and align the manuscript with the journal's standards, I think the editors will consider its acceptance.

Reviewer #2: We have no further comments.

Reviewer #3: The following aspects should be commented on: The duration of the the PPI use. It should be remembered that the Spanish family with missense of the K+H+-ATPase developed tumors (NETs in the twenties and cancer in the thirties)(Calvete et al). Could the apparent positive effect by the use of PPIs many years before the diagnosis of cancer be due to endoscopy at that time eliminating persons with precancer changes?. Every condition with long-term hypergastrinemia in animals gives gastric neoplasia. Even in man all other causes of long-term hypergastrinemia predisposes to gastric cancer. Eradication of H. pylori followed by PPI treatment causes without doubt gastric cancer and there are many papers describing gastric NETs secondary to PPI treatment, and even some case reports connecting gastric cancer to PPI use. All these aspects should be included in the discussion..

Requests from Editors:

GENERAL

* Please confirm that your title complies with to PLOS Medicine's style. Your title must be nondeclarative and not a question. It should begin with main concept if possible. "Effect of" should be used only if causality can be inferred, i.e., for an RCT. Please place the study design ("A randomized controlled trial," "A retrospective study," "A modelling study," etc.) in the subtitle (i.e., after a colon).

* Statistical reporting: Please revise throughout the manuscript, including tables and figures.

- Please report statistical information as follows to improve clarity for the reader ""XX (95% CI [XX,XX]; p</=)"".

- Please separate upper and lower bounds with commas instead of hyphens as the latter can be confused with reporting of negative values.

- Please repeat statistical definitions (HR, CI etc.) for each set of parentheses.

* Please ensure that all abbreviations are defined at first use throughout the text (including statistical abbreviations). Please also check figures and tables.

* Please ensure that tables and figures, including those in supplementary files, are appropriately referenced in the main text.

* Please check that any use of statistical terms (such as trend or significant) are supported by the data, and if not please remove them. Please note that the term trend should be used only when the test for trend has been conducted. 

* Please review your text for claims of novelty or primacy (e.g. 'for the first time') and remove this language.

* Your study is observational and therefore causality cannot be inferred. Please remove any language that implies causality, such as effect. Refer to associations instead.

* Although "risk" and "odds" are often used interchangeably in common conversation, they have very specific meanings in medical research. Since you are reporting odds ratios, we believe it is clearer to refer to "odds" throughout and remove the word "risk."

* Please note that you indicated in the online submission form that the study received no funding, yet you provided a financial disclosure statement after the main text. Please revise the online submission form to ensure that all statements are correct and updated according to our requirements.

* Please include the statement on code availability in the data availability statement.

ABSTRACT

* Please confirm that your abstract complies with our requirements, including providing all the information relevant to this study type https://journals.plos.org/plosmedicine/s/submission-guidelines#loc-abstract

* Please clarify what you mean by ‘ethnic sector’. Please also see our comments under 'Methods and Results'.

* Please provide the main baseline characteristics of the study population.

* In the abstract, please confirm that you included the important dependent variables that are adjusted for in the analyses.

* “we did not detect a harmful association between PPI use and upper GI cancer” – for transparency, please report the numerical values (at least an example).

METHODS AND RESULTS

* Upon further thought, we believe that the RECORD guideline/checklist might be a better fit for your study. Please ensure that the study is reported according to the RECORD guideline (available from https://www.record-statement.org) and include the completed checklist as Supporting Information. Please add the following statement, or similar, to the Methods: "This study is reported as per the Reporting of Studies Conducted using Observational Routinely-Collected Data (RECORD) guideline (S1 Checklist)." When completing the checklist, please use section and paragraph numbers, rather than page numbers.

* “ethnic sector (general population, Ultra-Orthodox Jewish, or Arab) – Could you please clarify how ethnicity/race was recorded and defined? How is the “general population” defined? Is "Ultra-Orthodox Jewish" an officially recognized ethnicity in Israel?

* “Ethnic sector classification was based on residential patterns and demographic characteristics, using algorithms validated in previous population based studies.” – please provide references. What are these algorithms? What does demographic characteristics exactly mean? Is ethnicity/race not recorded in EHRs?

* “socioeconomic status (SES), based on residential address and grouped into six ordinal categories” – please provide these categories here.

* For BMI, please provide a unit/definition.

* “Patient and public involvement” – We suggest removing this paragraph, but will leave it up to your discretion.

* “Reporting guideline” – We suggest adding the statement about the reporting guideline to another section of the Methods, such as "Study Design and Data Source."

* “Cancer cases” – Please replace ‘cases’ with participant, patient, individual, or person and ensure to use patient-centered language throughout your manuscript. Patient-centered language is constructed with the use of post-modified nouns (e.g. 'patients with cancer’ (or similar) instead of ‘cancer patients’) putting the person first in the sentence structure.

* Table 2: Please report confidence intervals.

* l.316, “Famotidine and calcium carbonate were not significantly associated with cancer risk.” – what about Lansoprazole?

* “alcoholism” – Please note that in the Methods section, you refer to alcohol use rather than alcoholism. We suggest using "alcohol use" or "alcohol consumption" throughout.

* Figure 3: Since you only have one graph, please note that the six diagnoses are listed under "Medication purchased at least once during the period," which seems inaccurate. We suggest splitting the graph.

* Table 1: Why aren't alcohol-related diagnoses or alcohol consumption presented in the baseline characteristics?

DISCUSSION

* “In conclusion, our findings do not provide evidence for a causal link between long term PPI use and upper gastrointestinal cancer.” – Please revise with regard to your study being observational.

* It seems that you did not have any data on medication adherence. Please address this as a limitation.

General Editorial Requests

---

## [Editor Report · Decision Letter 3]

31 Oct 2025

Dear Dr. Israel,

Thank you very much for re-submitting your manuscript "Association between Proton Pump Inhibitor Use and Upper Gastrointestinal Cancer: A Matched Case-Control Study Accounting for Reverse Causation and Confounding by Indication" (PMEDICINE-D-25-02605R3) for review by PLOS Medicine.

Thank you for your detailed response to the reviewers' and editor’s comments. There are a few minor editorial issues that need to be addressed before we can accept the manuscript for publication. When submitting your revised paper, please once again include a detailed point-by-point response to the editorial comments. Please revise the paper accordingly, and submit the final revision by November 7. The remaining issues that need to be addressed are listed at the end of this email.

In revising the manuscript for further consideration here, please ensure you address the specific points made by the editors. In your rebuttal letter you should indicate your response to the editors' comments and the changes you have made in the manuscript. Please submit a clean version of the paper as the main article file. A version with changes marked must also be uploaded as a marked up manuscript file. Please also check the guidelines for revised papers at http://journals.plos.org/plosmedicine/s/revising-your-manuscript for any that apply to your paper.

A reminder that when your manuscript is accepted, an uncorrected proof of your manuscript will be published online ahead of the final version, unless you've already opted out via the online submission form. If, for any reason, you do not want an earlier version of your manuscript published online or are unsure if you have already indicated as such, please let the journal staff know immediately at plosmedicine@plos.org.

Please do not hesitate to contact me directly with any questions (atosun@plos.org). If you reply directly to this message, please be sure to 'Reply All' so your message comes directly to our inbox.

We look forward to receiving the revised manuscript. 

Sincerely,

Alexandra Tosun, PhD

Senior Editor 

PLOS Medicine

plosmedicine.org

Requests from Editors:

* Title: We suggest changing the title to: Association between Proton Pump Inhibitor Use and Upper Gastrointestinal Cancer: A Matched Case-Control Study 

* Abstract: When reporting age, please add ‘year’ as unit.

* Table 2: Please add a heading for the CI values, such as ‘ 95% confidence intervals’.

* Discussion: “In conclusion, this observational study does not provide evidence for a causal link between long-term PPI use and upper gastrointestinal cancer. – Thank you for revising the sentence. Because of the way the study was designed, it would be impossible to show evidence of a causal link, which makes the statement unnecessary. We suggest changing it to: In conclusion, this study does not provide evidence for an association between long-term PPI use and upper gastrointestinal cancer.

* Thank you addressing our requests regarding the definition of ‘ethnic sector’. In your rebuttal, you have stated that you clarified the meaning of ethnic sector directly in the abstract and Methods. The paragraph you have provided in your response does not reflect the changes in the manuscript and the manuscript is still lacking a reference to the algorithms. Please add the statement from your rebuttal to the Methods Section, “Ethnic sector classification was based on residential clustering algorithms validated against census data in prior national studies[1]. The algorithm identifies ethnic sector based on neighborhood demographic composition; ethnicity is not directly recorded in Israeli EHRs."

* “as defined by the Israeli Central Bureau of Statistics”, “These groupings are consistent with prior national health research and allow for adequate discrimination across the socioeconomic gradient.” – we think it would be useful to add these explanations in the Methods Section.

* Please note that we have updated the data availability statement in the online submission form to include the statement on code availability. We have also updated the Financial Disclosure statement. Please check.

---

## [Editor Report · Decision Letter 4]

18 Nov 2025

Dear Dr Israel, 

On behalf of my colleagues and the Academic Editor, Gilaad G. Kaplan, I am pleased to inform you that we have agreed to publish your manuscript "Association between Proton Pump Inhibitor Use and Upper Gastrointestinal Cancer: A Matched Case-Control Study Accounting for Reverse Causation and Confounding by Indication" (PMEDICINE-D-25-02605R4) in PLOS Medicine.

I appreciate your thorough responses to the reviewers' and editors' comments throughout the editorial process. We look forward to publishing your paper.

PRESS

Thank you again for submitting to PLOS Medicine.

Sincerely, 

Alexandra Tosun, PhD 

Senior Editor 

PLOS Medicine